## [Reviewer comments · BMJ Open]

ARTICLE DETAILS

TITLE (PROVISIONAL)	Mindfulness-based programmes for mental health promotion in adults in non-clinical settings: protocol of an individual participant data meta-analysis of randomised controlled trials
AUTHORS	Galante, Julieta; Friedrich, Claire; Dalgleish, Tim; White, Ian; Jones, Peter

VERSION 1 – REVIEW

REVIEWER	Taylor, SL Veterans Health Administration
REVIEW RETURNED	12-Nov-2021

GENERAL COMMENTS	The authors were responsive to my comments. I look forward to seeing the results of this paper.
---

REVIEWER	Cristea, Ioana-Alina Universitatea Babeş-Bolyai Facultatea de Psihologie si Stiinte ale Educatiei, Clinical Psychology and Psychotherapy
REVIEW RETURNED	07-Dec-2021

GENERAL COMMENTS	The authors have addressed one of my main concerns by judiciously explaining why they are using the two-stage individual patient data meta-analysis model. However, they have maintained what is in my view a major limitation, which is selecting studies based on risk of bias and moreover on an arbitrary and unfounded consideration of a total score on the risk of bias tool. As I explained in my previous review, the risk of bias tool is not a scale, it is not validated as such, and a suite of papers I cited and several others have shown that at least some risk of bias domains are independent from each other. It is incorrect that my concern was based primarily on the difficulty in retrieving studies, my concern was based on the fact that this selection filter was inappropriate and will create a false equivalence between studies that have very different types of flaws. I add that the Cochrane Handbook specifically recommends in this exact matter: " (1) Primary analysis restricted to studies at low risk of bias The first approach involves restricting the primary analysis to studies judged to be at low risk of bias overall. Review authors who restrict their primary analysis in this way are encouraged to perform sensitivity analyses to show how conclusions might be affected if studies at a high risk of bias were included." (https://training.cochrane.org/handbook/current/chapter-07#section-7-6-2) However, the authors are clearly very advanced with this meta-analysis having already retrieved most IPD, so there is not much point in debating this further. Another point, there is no discussion of any data and code
--

	sharing. Even if IPD cannot be shared, aggregate data and analysis code definitely can be shared. Finally, stating results will be shared in a high-impact publication sounds a bit unnecessary (scientific publication would suffice).
--	---

VERSION 1 – AUTHOR RESPONSE

Reviewer: 1

Dr. SL Taylor, Veterans Health Administration, University of California Los Angeles Comments to the Author:

The authors were responsive to my comments. I look forward to seeing the results of this paper.

Reply: Thank you.

Reviewer: 2

Dr. Ioana-Alina Cristea, Universitatea Babes-Bolyai Facultatea de Psihologie si Stiinte ale Educatiei, Universitatea Babes-Bolyai Facultatea de Psihologie si Stiinte ale Educatiei Comments to the Author:

The authors have addressed one of my main concerns by judiciously explaining why they are using the two-stage individual patient data meta-analysis model. However, they have maintained what is in my view a major limitation, which is selecting studies based on risk of bias and moreover on an arbitrary and unfounded consideration of a total score on the risk of bias tool. As I explained in my previous review, the risk of bias tool is not a scale, it is not validated as such, and a suite of papers I cited and several others have shown that at least some risk of bias domains are independent from each other. It is incorrect that my concern was based primarily on the difficulty in retrieving studies, my concern was based on the fact that this selection filter was inappropriate and will create a false equivalence between studies that have very different types of flaws. I add that the Cochrane Handbook specifically recommends in this exact matter: " (1) Primary analysis restricted to studies at low risk of bias The first approach involves restricting the primary analysis to studies judged to be at low risk of bias overall. Review authors who restrict their primary analysis in this way are encouraged to perform sensitivity analyses to show how conclusions might be affected if studies at a high risk of bias were included." (<https://training.cochrane.org/handbook/current/chapter-07#section-7-6-2>) However, the authors are clearly very advanced with this meta-analysis having already retrieved most IPD, so there is not much point in debating this further.

Reply: We understand this concern and have reported this as a limitation. In the new version we have expanded on this limitation and described remedial actions: "We acknowledge, however, that this criterion is sub-optimal, potentially imposing a limitation on our findings. The RoB2 tool has not been validated as a scale, so there are no validated cut-off points and domains may not be interchangeable 48. Therefore, the included studies may have very different types of flaws. These flaws will be described through a detailed assessment of the risks of bias of each of the included studies using the RoB2 tool. We will also discuss our findings in relation to the sensitivity analysis performed in our previous review." (Risk of bias assessment section).

Unfortunately we cannot perform a sensitivity analysis including IPD from studies at higher risk of bias, as the Cochrane manual recommends, because that would involve collecting data from many more trials, which is unfeasible. However, our previous aggregate data meta-analysis could be taken as a basic type of sensitivity analysis about what happens if we include the higher risk studies.

Another point, there is no discussion of any data and code sharing. Even if IPD cannot be shared, aggregate data and analysis code definitely can be shared.

Reply: Thank you for reminding us of this. The new version states that aggregate data and analysis code will be shared in a public repository.

Finally, stating results will be shared in a high-impact publication sounds a bit unnecessary (scientific publication would suffice).

Reply: We have made this change.